# The Cardioprotective Properties of Selected Nuts: Their Functional Ingredients and Molecular Mechanisms

**DOI:** 10.3390/foods13020242

**Published:** 2024-01-11

**Authors:** Beata Olas

**Affiliations:** Department of General Biochemistry, Faculty of Biology and Environmental Protection, University of Lodz, Pomorska 141/3, 90-236 Lodz, Poland; beata.olas@biol.uni.lodz.pl; Tel./Fax: +48-42-6354485

**Keywords:** cardioprotective disease, nuts

## Abstract

Nuts have been known as a nutritious food since ancient times and can be considered part of our original diet: they are one of the few foods that have been eaten in the same form for thousands of years. They consist of various dry fruits and seeds, with the most common species being almonds (*Prunus dulcis*), hazelnuts (*Corylus avellana*), cashews (cashew nuts, *Anacardium occidentale*), pistachios (*Pistacia vera*), walnuts (Italian nuts, *Juglans regia*), peanuts (*Arachia hypogaca*), Brazil nuts (*Bartholletia excels*), pecans (*Corya illinoinensis*), macadamia nuts (*Macademia ternifolia*) and pine nuts. Both in vitro and in vivo studies have found nuts to possess a range of bioactive compounds with cardioprotective properties, and hence, their consumption may play a role in preventing and treating cardiovascular diseases (CVDs). The present work reviews the current state of knowledge regarding the functional ingredients of various nuts (almonds, Brazil nuts, cashew nuts, hazelnuts, macadamia nuts, peanuts, pecan nuts, pine nuts, pistachios, and walnuts) and the molecular mechanisms of their cardioprotective action. The data indicate that almonds, walnuts and pistachios are the best nut sources of bioactive ingredients with cardioprotective properties.

## 1. Introduction

Nuts have been known to man since ancient times. They have been eaten in the same form for thousands of years and can be considered part of our original diet. Nuts are typically described as dry fruits with an edible seed and a hard shell, with well-known examples being almonds (*Prunus dulcis*), hazelnuts (*Corylus avellana*), cashews (cashew nuts, *Anacardium occidentale*), pistachios (*Pistacia vera*), walnuts (Italian nuts, *Juglans regia*), peanuts (*Arachia hypogaca*), Brazil nuts (*Bartholletia excels*), pecans (*Corya illinoinensis*), macadamia nuts (*Macademia ternifolia*) and pine nuts. In general, there are four groups of nuts: (1) classic nuts in the botanical sense, such as walnuts: hard, dry, growing directly into a new plant; (2) drupes, such as almonds: the nut is a fruit pit; (3) seeds of gymnosperms, such as pine nuts: the seed is practically bare, typically covered with a hard husk and no soft matter; (4) angiosperm seeds, such as Brazil nuts and macadamia nuts: the nut is surrounded by a classic, large fruit. Hazelnuts are the second most popular nuts worldwide after almonds. However, walnuts are actually the oldest tree food known to man. Some researchers claim that humans have been eating walnuts since as far as 100,000 years ago. It is also interesting that peanuts are botanically defined as legumes; however, they have a similar nutrient composition and culinary use as tree nuts and are, therefore, usually included as nuts when estimating total nut intake [1,2,3,4].

All nuts contain large amounts of lipids. Furthermore, both in vitro and in vivo studies have found nuts to possess a range of bioactive compounds with cardioprotective action (Figure 1) [5,6,7,8,9,10,11,12]. Hence, there is great interest in their use as food products for the prevention and treatment of cardiovascular diseases (CVDs) [1,2,3,4]. Recent papers have reviewed the phytochemical characteristics and biological properties of mixed and selected nuts, for example, almonds, and their cardioprotective potential [1]. It is interesting that raw cashews contain urushiol, a toxin that causes a delayed allergic skin reaction similar to that caused by poison ivy. The poison is contained in the shells, so the kernels are perfectly safe [1,2,3,4].

The aim of the present review is to compare the functional ingredients of certain commonly used nuts with cardioprotective potential, viz. almonds, Brazil nuts, cashew nuts, hazelnuts, macadamia nuts, peanuts, pecan nuts, pine nuts, pistachios and walnuts, and summarize the molecular mechanisms behind them. It also indicates the nuts with the best sources of cardioprotective compounds, viz. almonds, walnuts, and pistachios.

The corpus of the review comprised papers identified in the Scopus, PubMed, Web of Knowledge, Science Direct and Web of Science electronic databases, among others. The last search was run on 19 December 2023. The following terms were used: “nuts”, “cardioprotective action” and “cardiovascular disease”. The search was restricted to English-language publications.

## 2. Functional Ingredients of Nuts

Nuts provide 23 to 30 kJ of energy per gram [3]. They are also a source of a range of chemical compounds, many of which have cardioprotective potential, such as polyunsaturated fatty acids (PUFA), phytosterols, phenolic compounds and fiber [13,14,15,16].

In walnuts, the lipid profile is mostly dominated by PUFAs, with linoleic and linolenic acids being the greatest contributors [3,4]. It is also well known that hazelnut PUFAs may benefit the serum lipid profile, mainly by protecting low-density lipoprotein (LDL) from oxidation and decreasing oxidized plasma LDL concentration [17]. Moreover, Durak et al. [18] report that hazelnut supplementation results in significantly lower plasma lipid peroxidation, indicated by malondialdehyde (MDA) level, and higher antioxidant potential.

Nuts also contain various saturated fatty acids (SFAs) and monounsaturated fatty acids (MUFAs). For example, a study of the total fatty acid content in Cosford, Webba Cenny and Katalonski hazelnuts by Ciemniewska and Ratusz [19] found the predominant fatty acid to be oleic acid in the form of MUFA, which was present in 80.25 g/100 g of the total fatty acids in the Webba Cenny variety. Peanuts, too, have high levels of MUFA (24.4 g/100 g), as noted by Mingrou et al. [20], and higher levels have been reported in macadamia nuts (58.9 g/100 g) and pecan nuts (40.8 g/100 g), respectively. It is interesting that Brazil nuts, cashews and macadamia nuts have high levels of saturated fatty acids that may be a risk factor in CVDs [1,2,3,4].

In addition, chemical changes have been observed in hazelnuts during their development, with the amounts of MUFAs and vitamin E increasing and PUFAs decreasing during kernel maturation [21,22,23]. Ciemniewska-Zytkiewicz et al. [23] noted the same changes in Katalonski hazelnuts grown in Poland.

Recently, sphingolipids, including sphingosine-1-phosphate (a bioactive metabolite that regulates diverse biological processes by binding to a family of G protein-coupled receptors or as an intracellular second messenger), were discovered in almonds and pistachios [3,4].

Importantly, nuts bestow cardioprotective benefits, which may be associated with their fiber content. Dietary fiber is believed to lower blood cholesterol levels by binding cholesterol and bile acids in the intestinal lumen. High-fiber diets also play important roles in preventing obesity. Fiber can absorb water and swell; this, combined with the ability to exchange cations, allows it to buffer and bind excess hydrochloric acid in the stomach, increase intestinal filling and stimulate peristalsis. It also creates a favorable substrate for the proper growth of microflora in the large intestine [1,2,3,4]. The nuts with the highest fiber content are believed to be almonds.

Some papers indicate that nuts are also plentiful sources of vitamins, including ascorbic acid, B_1_, B_2_, B_3_, B_5_, B_6_ and B_9_, carotenoids, including vitamin A (almond—2 µg/100 g; walnut—21 µg/mL; pecan—55 µg/mL; and hazelnut—106 µg/100 g), and vitamin E. Peanuts, walnuts, almonds, pistachios, and cashew nuts are all abundant source of B vitamins [1,2,3,4,16].

Vitamin E, consisting of α-, β-, γ- and δ-tocopherols, is a fat-soluble vitamin that exerts potent antioxidant activity by chain-breaking reactions occurring during unsaturated lipid peroxidation [24,25,26,27]. Vitamin E is also known to protect LDL against oxidation. In addition, α-tocopherol demonstrates considerable protection against coronary artery disease.

Like almonds, hazelnuts demonstrate high levels of α-tocopherol (31.4 mg/100 g of kernel oil), as well as β- and γ-tocopherol (6.9 mg/100 g of oil), and δ-tocopherol (0.1 mg/100 g of oil) [28]. Ciemniewska-Zytkiewicz et al. [29] found α-tocopherol to be the most abundant tocopherol (90–92% of total tocopherol content) in two hazelnut cultivars grown in Poland, with bound tocopherols representing 45.5% of total tocopherol content in the Katalonski cultivar and 21.7% in Webba Cenny.

Nuts are a good source of minerals, which can also decrease the risk of cardiovascular diseases. For example, increasing potassium intake with food, i.e., about 3500 mg/day among adults, has been found to lower blood pressure. The potassium content of pistachios is 642–1025 mg/100 g of nuts and almonds 728 mg/100 g. Nuts are also good sources of L-arginine, a key component of nitric oxide (NO), which acts as an important endogenous vasodilator and regulator of blood pressure [3,4].

Nuts also contain high levels of phytosterols. Phytosterol is a collective term encompassing both unsaturated and saturated sterols, possessing an alkane bond at the C_1_ position on the B ring. They interfere with cholesterol absorption and help lower blood cholesterol levels. Phytosterols can also reduce the level of SFA and have been found to have antioxidant activity [30,31]. The highest levels of phytosterols among nuts are found in pistachios (272 g/100 g) [3,4]; however, almonds are also good sources of phytosterols. For example, a study of seven almond varieties found phytosterol content to range between 103 and 206 g/100 g [32]. It has also been found that 70–80% of phytosterols in roasted peanuts are β-sitosterol [33]. In addition, Cherif et al. [34] isolated and identified five triterpenes, which are synthesized via the same pathway as phytosterols.

Phenolic compounds have also been found to be present in nuts, with each species having a characteristic phenolic compound profile [35]. For example, Tas and Gokmen [36] found high levels of procyanidins A, B, trimers and tetramers, and prodelphinidin in peanuts. Peanut skins contain about 17% procyanidins, composed of a mixture of low- and high-molecular-weight oligomers [37]. Smeriglio et al. [38] report high levels of isorhamnetic, kaempferol, and quercetin in almonds; despite not being an essential part of the human diet, their consumption offers a range of health benefits.

Importantly, total phenolic compound content (typically 291–835 mg/100 g [25]) has a significant effect on hazelnut quality [26,39]. The most abundant phenolic compounds in hazelnuts are gallic acid, catechin, and epicatechin gallate [40,41]. Hazelnuts contain high levels of phenolic acids (2 mg/100 g [42]) and flavonoids (12 mg/100 g [25]), together with flavonols in the skin and shells. The primary flavonoids are catechins and gallocatechins [25]. The main phenolic acid in hazelnut shells is gallic acid (3,4,5-trihydroxybenzoic acid), which has anti-cancer, anti-inflammatory and antioxidant properties, while the antioxidant flavan-3-ol is present in the kernels.

More than 60% of the phenolic compounds identified in hazelnuts are tannins. Although polymeric forms predominate, most are condensed down to 10-degree oligomers. These polymers reduce reactive oxygen species (ROS) production and chelate metal ions, particularly copper and iron [29,39,43,44,45,46,47].

In a study of various nuts, Gu et al. [48] found the highest concentrations of proanthocyanidins to be present in hazelnuts (491 mg/100 g). A-type proanthocyanidins have also been found in hazelnuts, with these being highly polymerized [25].

Wozniak et al. [49] used a combination of ultra-high pressure liquid chromatography (UPLC) and atomic absorption spectrometry to study the phenolic compounds and mineral content in peanuts and nine tree nuts commonly available on the Polish market: pecans, cashews, walnuts, macadamia nuts, almonds, hazelnuts, Brazil nuts, pine nuts and pistachios. They found the highest total flavonoid content in walnuts (114.861 µg/g) and the lowest in almonds (1.717 µg/g). The most abundant flavonoid in most of the tested nuts was epicatechin; however, it was absent from almonds. The highest content of trace elements was determined in pine nuts (192.79 µg/g) and the lowest in pistachios (93.24 µg/g).

The total phenolic compound content and antioxidant properties of hazelnut extract are also influenced by the choice of solvent and time of contact [46]. The highest total phenolic compound content was obtained by boiling water and 80% (*v*/*v*) aqueous acetone, while the highest antioxidant activity was obtained by contact with 80% (*v*/*v*) aqueous acetone for 24 h; the latter was probably related to the higher concentration of antioxidant compounds, such as tocopherols [43] and tannins, particularly condensed tannins [26,27]. In addition, the extracts of almonds, pine nuts and peanuts produced under these conditions were found to have lower antioxidant activity than the hazelnut extract [46].

Another set of phenolic compounds present in *C. avellana* are diarylheptanoids, i.e., with a 1,7-diphenylheptane structural skeleton; these have been observed in various parts, including the shells [50,51,52,53,54,55]. Their structure generally consists of two aromatic rings connected by a seven-membered aliphatic chain [56]. The chemical contents of various nuts are reviewed in more detail elsewhere [4,25,57].

In addition, various stilbenoids, i.e., other phenolic compounds, have been isolated from peanut seeds: resveratrol, *trans*-arachidin 1,3: *trans*-arachidin 2,4: *trans*-arachidin 3,5: isopentadienylresveratrol [58,59,60].

Interestingly, nuts are also good sources of selenium, which can help prevent the onset of cardiovascular disease. Brazil nuts are a particularly good source [61,62]. Moreover, a 30 g serving of cashew nuts has around 30% of the recommended daily intake of copper. Hazelnuts, Brazil nuts, walnuts, pecans and pine nuts are also good alternative options [3,4]. In addition, macadamia nuts, pine nuts, and cashews are sources of iron [3,4]. It is interesting that Brazil nuts (26% of Recommended Dietary Allowances (RDA)), cashews (20% of RDA), almonds (19% of RDA), and pistachios (8% of RDA) are good sources of magnesium [3,4]. Zinc was also found in various nuts, including pine nuts (8.7 mg/1 cup), hazelnuts (5.7 mg/1 cup), almonds (4.5 mg/1 cup), pecans (4.9 mg/1 cup), walnuts (3.6 mg/1 cup), and pistachios (2.9 mg/1 cup) [3,4].

It is interesting that pistachios contain chlorophyll, which is where they get their green color. In a way, they are the nutty equivalent of kale [1,2,3,4].

Table 1 presents the chemical content of the selected nuts and highlights those with the highest levels of cardioprotective compounds, including PUFAs, phytosterols, fiber, phenolic compounds and selenium. These compounds are particularly plentiful in almonds, pine nuts, Brazil nuts, pistachios, walnuts and hazelnuts (≥2 compounds). However, almonds, walnuts and pistachios are better sources of cardioprotective PUFAs, phytosterols, selenium and phenolic compounds than pine nuts.

Nuts can be processed in various ways to create different final products. However, while it is well known that the choice of processing method can affect the bioavailability of the obtained nut compounds, the precise impact of each method remains unclear. Nuts can be consumed peeled, blanched, roasted or deshelled. They are often consumed roasted, which enhances their flavor, color and crunchiness [13,57,63,64]. While roasting is commonly used to preserve the quality and storability of nuts, the process alters their microstructure and chemical content, resulting in color changes and lipid modification. In addition, roasting may decrease the antioxidant properties in certain nuts, including hazelnuts and walnuts, but can maintain or even enhance them in almonds and pistachios [57,65]. Matoes et al. [66] suggest that this effect may be associated with a decrease in phenolic compound content. However, the impact of roasting on phenolic compounds in nuts remains unclear [67,68,69,70]. Some studies indicate that their content may be decreased by higher temperatures and longer heating and decreased by lower temperatures or shorter heating times [71]. For example, phenolic components (total phenols, flavonoids, condensed tannins, and phenolic acids) of almond kernels were substantially lost in the initial phase; afterward, these components gradually increased with roasting temperature and duration. Similar results were also observed for their antioxidant activities [71]. Barreca et al. [2] report that roasting induces chemical and microstructural changes in almonds, favoring lipid oxidation and modulating antioxidant compound content; it also alters the protein profile and allergenic properties of nuts in general [72]. Interestingly, hot water blanching also influences the protein composition of nuts; however, the effect depends on the species and conditions of processing [73].

## 3. Nuts as a Key Food Component in Cardioprotection

Various international health organizations, including the American Heart Association (2021), Canadian Cardiovascular Society (2016), European Society of Cardiology and European Atherosclerosis Society (2019), emphasize the value of nuts for primary and secondary prevention of cardiovascular risk; however, the mechanisms by which they exert these effects remain unknown [74]. The organizations recommend nuts as a source of healthy fats, such as unsaturated fatty acids and phytosterols, plant protein, fiber, and various minerals, such as potassium and phenolic compounds that can lower LDL cholesterol. Consumption is also believed to improve the overall lipoprotein profile and decrease various other risks associated with CVD. However, Food Standards Australia New Zealand (FSANZ) indicate that 30 g per day of walnuts may improve endothelium-dependent vasodilation [74].

### 3.1. Mixed Nuts

Various studies in animal and human models indicate that mixed nuts have cardioprotective potential. For example, in a model of male mice fed with an atherogenic diet, supplementation with a nut mixture comprising peanuts, macadamia nuts, almonds, pistachios, walnuts, Brazil nuts and cashews resulted in a lower concentration of oxLDL compared to control apoE-knockout mice [75].

Arnesen et al. [76] found nut consumption to be associated with a lower risk of coronary heart disease, possibly through their influence on blood lipids. Houston et al. [77] also report that mixed nut consumption can reduce the risk of CVD, most likely through a range of multifactorial effects.

A systematic review and meta-analysis of randomized controlled trials by Abbasifard et al. [78] demonstrated that mixed nuts significantly decrease serum oxLDL. A meta-regression found a significant correlation between nut type and oxLDL level, with pistachios having the most significant effect on reducing circulating oxLDL; this was also noted in apoE-deficient mice receiving supplementation (3%) of mixed walnuts, hazelnuts and almonds. Another systematic review and meta-analysis indicate that adding a mixture of walnuts, cashews, almonds, pistachios and peanuts (excluding nut oil) into the regular diet reduces blood levels of apoliporotein B and improves high-density lipoprotein (HDL) function. The effect was not modified by sex, age or nut processing method; however, a lower body mass index (BMI) and higher baseline lipid concentrations were both found to enhance blood lipid/lipoprotein responses [79].

A systematic review and meta-analysis of 19 prospective cohort studies by Becerra-Tomas et al. [80] found mixed nut consumption to have a negative correlation with CVD incidence, CVD mortality, stroke mortality, coronary heart disease incidence and mortality. Similarly, Schwingshackl et al. [81] and Nora et al. [82] both reported associations between mixed nut consumption and a reduction in incident hypertension. Interestingly, Glenn et al. [74] also note that consumption of mixed nuts appears to influence the risk of various CVDs. For example, higher consumption was associated with a reduced risk of various CVDs, including a 24% lower risk of coronary heart disease and a 27% reduction in mortality; it was also associated with a 15% lower risk of atrial fibrillation and an 18% lower risk of stroke mortality.

In addition, various systematic reviews and meta-analyses have highlighted that medium nut consumption has cardioprotective potential [83,84,85,86,87,88,89,90]. In contrast, in a randomized clinical trial of 60 patients with metabolic syndrome, six-week supplementation with 30 g/day of mixed nuts (15 g walnuts, 7.5 g pine nuts, and 7.5 g roasted peanuts) did not appear to result in any changes in oxidative stress markers [91]. However, Lopez-Uriarte et al. [92] report that 12-week consumption of mixed nuts (15 g walnuts, 7.5 g almonds, and 7.5 g hazelnuts) reduced oxidative stress in patients with metabolic syndrome (n = 50). A recent study also found that the consumption of 1.5 oz mixed tree nuts per day (for 12 weeks) affected tryptophan metabolism, a factor in CVDs, in 56 overweight and obese subjects [93].

A recent study examined the effects of mixed nut consumption (42.5 g/day; cashews, macadamia nuts, Brazil nuts, almonds, pistachios, pecans, walnuts, and peanuts) for 16 weeks on LDL cholesterol, lipoprotein(a), and other cardiometabolic risk factors in overweight and obese adults (n = 34, 20–55 years, BMI = 25–40 kg/m^2^) [82]. The results indicate that consumption appeared to influence LDL cholesterol levels, and the participants demonstrated significantly lower body fat percentage, higher adiponectin, and lower diastolic blood pressure. It was also associated with a non-significant rise in total antioxidant capacity (TAC).

Park et al. [94] studied the relationship between mixed nut consumption (peanuts, pine nuts, and almonds, defining 15 g as one serving dose) and left ventricular hypertrophy in women (N = 12.257). The frequency of nut supplementation was categorized into five groups: four times a week or more, two to four times a week, once or twice a week, between once a month and once a week, and less than once a month. It was found that consuming nuts at least once per week was associated with a decreased probability of left ventricular hypertrophy.

A review by Weschenfelder et al. [95] found nut consumption to have various beneficial effects on glucose control and appetite suppression. In addition, the fatty acid, phytosterol, fiber and L-arginine content positively influenced the activity of adipokines and various metabolites related to adipose tissue and gut microbiota.

Figure 2 summarizes the associations between mixed nut consumption and CVD risk factors reported in epidemiologic studies and randomized clinical trials.

### 3.2. Almonds

Various authors have noted that regular almond consumption appears to have beneficial effects on the prophylaxis and treatment of CVDs. For example, almond consumption was associated with a reduction in both LDL cholesterol and total cholesterol levels; however, no significant effect was observed on HDL cholesterol, triglyceride levels, or LDL/HDL ratio [96]. Similar results were obtained following six-week supplementation with 20 g day^−1^ almonds in subjects with mild hypercholesterolemia, with reductions observed in LDL cholesterol and total cholesterol [97]. Similar effects were noted in hyperlipidemic subjects supplemented with 10 mL almond oil twice daily for four weeks [98]. Hyson et al. [99] also found almonds and almond oil to have beneficial effects on plasma lipid and LDL oxidation in healthy men and women (n = 22).

Moosavian et al. [100] observed that almond intake (≥50 g/day) results in a significant reduction in LDL in patients with type 2 diabetes. However, no significant effect was observed on total cholesterol, HDL or triglyceride levels, or on BMI or blood pressure.

The effect of almond consumption in obese and overweight subjects varies considerably between studies [101,102]. Dhillon et al. [102] propose that the high unsaturated fatty acid content in almonds, with high fat oxidation rates, contributes to a reduction in visceral fat. In this clinical trial, 86 healthy subjects (BMI 25–40 kg/m^2^) consumed an almond-enriched hypocaloric diet for 12 weeks. In contrast, ref. [101] report that a very long-term diet lasting 18 months (28 almonds daily) had no significant effect on weight loss, body composition or blood pressure in obese subjects; however, significant reductions in total cholesterol and triglyceride levels were noted.

Some authors attribute the beneficial effect of almond supplementation on blood lipid profile to their chemical composition, particularly its unsaturated fatty acid, phytosterol and phenolic compound levels, including flavonoids. This effect may be strongly influenced by the anti-inflammatory potential of almonds, bestowed primarily by their high MUFA content [103]. However, Rajaram et al. [103] indicate that almond consumption has no effect on various markers of inflammation, including E-selectin and C-reactive protein (CRP) in healthy adults (n = 25, age 22–53 years).

Interestingly, supplementation with 84 g or 168 g almonds daily for four weeks was found to have antioxidant activity in 30 young habitual smokers consuming 10 to 20 cigarettes per day [104,105]; consumption was found to reduce the levels of various parameters of oxidative stress, including MDA [104]. However, Chen et al. [106] found 85 g/day almond consumption for 22 weeks to have no significant effect on biomarkers of oxidative stress in patients with coronary artery disease (n = 45).

### 3.3. Pistachios

Various studies suggest that pistachio nuts appear to have cardioprotective properties. For example, Lippi et al. [107] report that consumption may have beneficial effects on blood lipid profile. In addition, research indicates that pistachio supplementation decreased total cholesterol concentration and LDL/HDL ratio in both healthy participants and patients with moderate hypercholesterolemia [108,109,110,111,112]. LDL concentration also decreased significantly in these groups, whereas others observed (LDL/HDL ratio) a non-significant reduction. Moreover, Sheridan et al. [110] found pistachio consumption to be associated with a significant increase in HDL concentration.

Pistachio consumption has also been repeatedly demonstrated to reduce both diastolic and systolic blood pressure [113,114,115,116]. It has also been found to improve some markers of inflammation, including serum interleukin 6, in young men [112]. Similar anti-inflammatory effects have also been noted in a rat model of ulcerative colitis [117]. Such inflammatory processes are also known to play an important role in the development of CVDs.

The antioxidant properties of pistachios have been demonstrated in vitro and in vivo in both animals and humans. A study carried out on 44 healthy volunteers found pistachio consumption to increase blood antioxidant potential compared to controls receiving a diet without nuts [109]. Pistachios were also found to have antioxidant potential in patients with metabolic syndrome, as indicated by measuring thiobarbituric acid reactive substances (TBARS), a marker of lipid peroxidation [118]. Additionally, eight-week pistachio supplementation reduced lipid peroxidation, as measured by using MDA, in hyperlipidemic rats fed a fat-rich diet [119]; however, supplementation did not result in a significant increase in superoxide dismutase (SOD) activity.

In addition, Paternili et al. [120] found pistachios to possess antioxidant potential in the J774 macrophage/monocyte cell line in vitro. Treatment with pistachio shells or roasted and salted pistachios in various concentrations reduced oxidative stress based on MDA assay and ROS production.

Recently, Ersoz et al. [121] found pistachio hull extract (50 and 100 mg/kg) to have antioxidant activity and to reduce apoptosis and DNA damage in Wistar albino rats (n = 12) with doxorubicin-induced cardiac damage.

Sehaki et al. [122] present more detail about the biological properties of pistachios which they attribute to the phenolic composition of these nuts.

### 3.4. Walnuts

Walnuts are also very good sources of compounds with cardioprotective activity. One study by Berryman et al. [123] evaluated the effect of the consumption of walnut skin (95.6 g), defatted walnut pulp (34 g), walnut oil (52 g) and whole walnuts (85 g) on oxidative stress in overweight or obese individuals with moderate hypercholesterolemia. An increase in the antioxidant potential was noted at all meals, as indicated by using the ferric reducing antioxidant power (FRAP) assay; however, a lower increase was noted for defatted walnut pulp.

It is important to note that walnut oil (40 µL/mL of cells–319 µL/mL of cells; 24 to 72 h) had positive effects on SOD activity in U937 cells (pro-monocytic, human myeloid leukaemia cell line) in vitro. Moreover, defatted walnut flour was found to have antioxidant effects on nerve cells following peroxide injury by preventing ROS production. In addition, the phenolic-rich walnut extract inhibited copper-mediated LDL oxidation by 84% in human plasma in vitro, compared to 14% by ellagic acid [124].

### 3.5. Other Selected Nuts

Pecan oil also appears to have antioxidant properties. In one experiment, male rats received a high-fat diet supplemented with either phenolic compound extract, pecan oil, or part or whole pieces of pecans for nine weeks; the greatest increase in antioxidant enzyme activity was observed in the group supplemented with pecan oil [125]. Recently, Cogan et al. [126] demonstrated that a pecan-enriched diet (66 g/day for four weeks) improved cholesterol profiles and enhanced postprandial microvascular reactivity in older adults (n = 44).

Brazil nuts have demonstrated high total antioxidant capacity in vitro [127], and consumption of one unit of nuts per day for three months was found to reduce oxidative stress in hemodialysis patients [128].

Duarte et al. [129] observed increased glutathione peroxidase activity in obese women who consumed one Brazil nut/day for two months, while Maranhao et al. [130] note a significant decrease in oxLDL in obese adolescents supplemented with 15–25 g of Brazil nuts per day for 16 weeks. In addition, Hughenin et al. [131] report decreased LDL oxidation in hypertensive and hyperlipidemic patients, and Colpo et al. [132,133] indicate significantly decreased total cholesterol and blood glucose levels in 130 healthy volunteers following Brazil nut consumption.

A systematic review found that regular intake (≥5 times per week) of 50–100 g of Brazil nuts may significantly decrease LDL and total cholesterol in healthy and hyperlipidemic subjects [134]. Maranhao et al. [130] also found three to five units of Brazil nuts per day to have cardioprotective effects in 17 obese women.

Some authors suggest that Brazil nut consumption may exert its cardioprotective effects by decreasing inflammatory parameters such as IL-1, Il-6, interferon-γ (INF-γ) and tumor necrosis factor-α (TNF-α), and by reducing oxidative stress [132,133,135]. More details about these mechanisms are presented by Ferrari [127].

A systematic review and a meta-analysis found that cashew nut supplementation did not appear to significantly influence the serum lipid profile, including HDL, LDL and total cholesterol levels, in 531 participants [136]. Similar results were obtained in another study with 392 participants, although a significant reduction was noted in systolic blood pressure [137].

Recently, Parilli-Moser et al. [138] found that regular consumption of peanuts seems to modulate lipid metabolism, reducing triglyceride blood concentrations. In this study, 63 healthy subjects consumed 25 g/day of skin roasted peanuts or peanut butter.

In addition, daily consumption of macadamia nuts did not appear to increase weight or body fat in overweight or obese adults, nor did it influence the level of cholesterol [139].

The cardioprotective potential of nut consumption is summarized in Figure 3, together with its mechanisms associated with the modulation of inter alia blood lipids, inflammation and oxidative stress. For example, oxidative stress has been found to trigger inflammation that can contribute to the development of CVDs. Such stress increases the activation of nuclear factor kappa beta (NFκβ), resulting in the release of pro-inflammatory cytokines and the inhibition of nuclear factor erythroid factor 2-related factor 2 (Nrf2), resulting in greater inflammation. Nut antioxidants, including phenolic compounds, quench free radicals, thus preventing the oxidative modification of LDL (oxLDL). This process prevents foam cell formation following macrophage oxLDL endocytosis.

Increased levels of free radicals in the sub-endothelial space stimulate changes in smooth muscle cell (SMC) phenotype, favoring the uptake of oxLDL. Such uptake induces inflammation, resulting in the synthesis of inflammatory cytokines, such as IL-6 and TNF-α, which promote atherosclerosis.

Moreover, the phytosterols, PUFAs and fiber present in nuts can improve lipid metabolism by lowering LDL cholesterol. Nut antioxidants can also quench the production of ROS in the cell, reduce oxidative stress, and suppress NFκβ expression and downstream pro-inflammatory cytokine production. These antioxidants may act as cofactors of various antioxidant enzymes [140]. However, more research is needed to determine the possible effects of nut consumption on both oxidative stress and inflammation, both of which play key roles in the development of CVDs.

## 4. Conclusions

The effect of nut consumption on various factors related to the development of CVDs has been investigated with regard to population, duration of treatment and intervention type; however, these studies have provided little specific data regarding the positive role of mixed nut consumption in CVDs. While some clinical trials have studied the effect of selected nuts, especially walnuts and almonds, on CVD and metabolic syndrome, few papers have evaluated the efficiency of less popular nuts such as pistachios, Brazil nuts and pecans in vivo. There is a clear need for more clinical trials investigating their cardioprotective potential. The phytochemical profiles of these less-known nuts are nonetheless interesting, and their consumption may play an important role in the prevention and treatment of CVDs; Brazil nuts, for example, are especially rich in selenium.

Although the present study provides an overview of the effects of the consumption of mixed and selected nuts on the prevention and treatment of CVDs (Table 2), the original studies employ a range of different standards for phytocompounds or nut products, and it is difficult to compare their findings. Furthermore, the cardioprotective potential of nuts appears to be influenced by numerous other factors, such as experimental model and dosage, and no specific clinical evidence exists regarding the efficacy, absorption and bioavailability of the various components of nuts.

## Figures and Tables

**Figure 1 foods-13-00242-f001:**
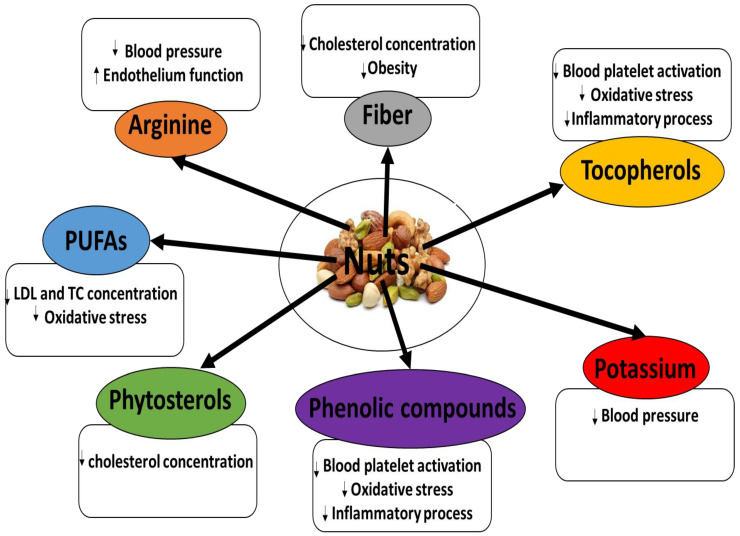
Key functional ingredients with cardioprotective potential present in nuts.

**Figure 2 foods-13-00242-f002:**
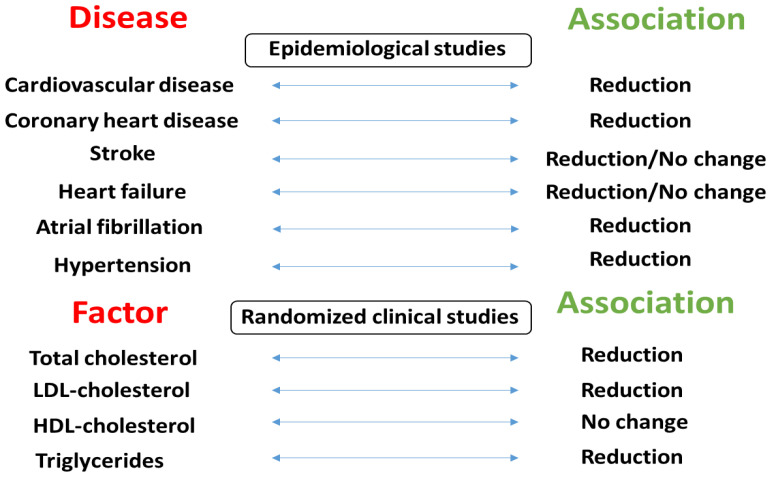
The relationships between mixed nut consumption and CVD risk factors observed in epidemiologic studies and randomized clinical trials.

**Figure 3 foods-13-00242-f003:**
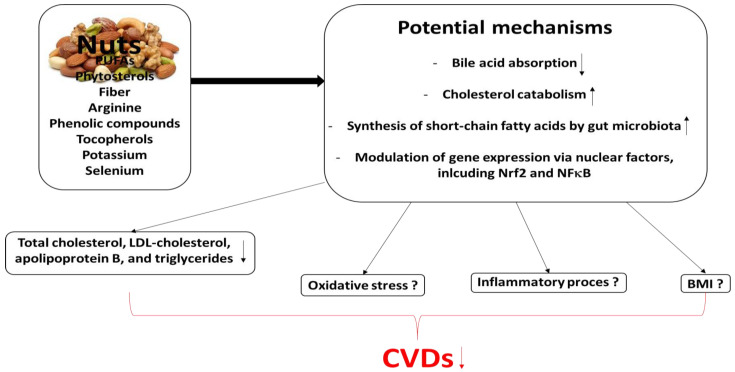
Potential mechanisms by which nut consumption may play a beneficial role in CVDs ([11]; modified).

**Table 1 foods-13-00242-t001:** Chemical content of selected nuts (compilation of data: [3,4]). Green fields indicate chemical compounds with cardioprotective potential. Grey fields indicate nuts with the highest concentrations of chemical compounds with cardioprotective properties.

Nuts	Protein (g/100 g)	Lipid (g/100 g)	SFA (g/100 g)	MUFA (g/100 g)	PUFA (g/100 g)	Phytosterols (g/100 g)	Fiber (g/100 g)	Phenolic Compounds (mg/100 g)	Selenium (µg/100 g)
**Almonds**	16.8–25.4	43.3–50.6	3.9	31.5	12.3	162	11.8–13.0	287	4.1–53.1
**Brazil nuts**	14.3	66.4	15.1	24.5	24.4	72	7.5	244	50–250
**Cashew nuts**	17.5–19.0	42.8–43.9	46.4	27.3	7.8	120	1.4–3.3	233	5.97–19.9
**Hazelnuts**	14.5–15.2	59.8–61.5	60.8	45.7	7.8	115	3.3–9.7	671	2.4–186
**Macadamia nuts**	7.5–8.6	76.0	76.0	58.9	1.4	119	8.0		3.6
**Peanuts**	25.8	49.2	6.2	24.4	15.6	126	8.5	126	7.2–9.3
**Pecan nuts**	9.0–9.3	72.0	6.2	40.8	21.6	113	9.6	406	1.14–3.8
**Pine nuts**	13.7	68.4	4.9	18.8	34.1	120	3.7	1284	0.7
**Pistachios**	19.4–22.1	44.4–45.4	5.4	25.0	14.1	272	10.3	58	7.0–89.3
**Walnuts**	14.4–16.0	64.5–65.4	6.1	8.9	38.1	143	6.7	1420	4.9

**Table 2 foods-13-00242-t002:** The effect of nut consumption on various parameters which may play an important role in the prevention and treatment of CVDs (in vivo human models).

Nuts	Sample Size, Duration of Intervention, and Dose	Characteristics of Participants	Results	References
**Mixed nuts**				
	583 participants, 3–8 weeks, 34–100 g/day	Normalipidemia and hypercholesterolemia	Decrease in LDL concentration	[141]
	2211 participants, 3 weeks-18 months, 30–85.5 g/day	Healthy or dyslipidemia	Decrease in triglyceride concentration	[142]
	2582 participants, 3–26 weeks, 5–100 g/day	Healthy adults	Decrease in triglyceride, LDL, and ApoB concentration	[85]
	1677 participants, 3–24 weeks, 15–168 g/day	Healthy adults	Decrease in triglyceride and LDL concentration	[70]
	1041 participants, 6–52 weeks, 6–128 g/day	Diabetic patients	Decrease in triglyceride concentration	[143]
	711 participants, 4–72 weeks, 20–60 g/day	Healthy adults	Decrease in triglyceride concentration	[144]
	60 participants, 6 weeks, 30 g/day	Metabolic syndrome	No change in ox-LDL and MDA	[91]
	50 participants, 12 weeks, 30 g/day	Metabolic syndrome	Reduction in oxidative stress	[92]
	56 participants, 12 weeks, 1.5 oz	Overweight and obese subjects	Changes in tryptophan metabolism	[93]
	34 participants, 16 weeks, 42.5 g/day	Overweight and obese subjects	Decrease in LDL concentration and diastolic blood pressure No change in oxidative stress	[82]
**Almonds**				
	142 participants, 4 weeks, 25–168 g/day	Healthy adults	Decrease in LDL concentration	[96]
	534 participants, 4–16 weeks, 37–113 g/day	Healthy adults	Decrease in triglyceride and LDL concentration	[145]
	2049 participants, 3–77 weeks, 10–168 g/day	Healthy adults	Decrease in triglyceride and LDL concentration	[146]
	120 participants, 3–12 weeks, 30–60 g/day	Diabetic patients	No change in lipid profile	[113]
	264 participants, 4–12 weeks, 29–113 g/day	Diabetic patients	Decrease in LDL concentration	[100]
	20 participants, 6 weeks, 20 g/day	Mildly hypercholesterolemic subjects	Decrease in total cholesterol and LDL	[97]
	85 participants, 20 weeks, 56 g/day	Healthy adults	Decrease in triglyceride, LDL and total cholesterol concentration	[68]
	30 participants, 4 weeks, 60 g/day	Mild hypercholesterolemia	Decrease in total cholesterol and LDL	[147]
	27 participants, 4 weeks, 73 g/day	Hyperlipidemic patients	Decrease in LDL, ox-LDL, lipoprotein A, and LDL/HDL ratio	[148]
	45 participants, 6 weeks, 85 g/day	Patients with coronary artery disease	No change in lipid profile and blood pressure; increased NO	[106]
	123 participants, 18 months, 28 g/day	Overweight or obese adults	Reduction in bodyweight, and no significant changes in body composition	[102]
	97 participants, 4 weeks, 10 mL oil twice daily	Hyperlipidemic patients	Decrease in total cholesterol and LDL	[98]
	30 participants, 4 weeks, 86 and 164 g/day	Male smokers	Decrease in MDA; increase in SOD and GSH-Px	[105]
	60 participants, 4 weeks, 84 g/day	Smokers	Decrease in MDA; no changes in SOD and GSH-Px	[104]
**Walnuts**				
	365 participants, 4–24 weeks, 15–108 g/day	Healthy adults	Decrease in triglyceride and LDL concentration	[149]
	1059 participants, 4 weeks-1 year, 15–108 g/day	Healthy adults	Decrease in triglyceride and total cholesterol concentration	[10]
	506 participants, 4–112 days, 30–108 g/day	Metabolic syndrome	Decrease in triglyceride concentration	[150]
	2466 participants, 4 weeks-2 years, 19.3–75 g/day	Healthy middle-aged and older adults	Decrease in triglyceride and total cholesterol concentration	[151]
**Cashews**				
	392 participants, 4–2 weeks, 30–42 g/day	Healthy adults	No change in lipid profile	[137]
**Pistachios**				
	771 participants, 3–24 weeks, 32–126 g/day	Healthy adults	Decrease in triglyceride, LDL and total cholesterol concentration	[152]
**Peanuts**				
	643 participants, 2–24 weeks, 25–200 g/day	Healthy adults	Decrease in triglyceride concentration	[138]
**Hazelnuts**				
	60 participants, 8 weeks, 15–30 g/day	Children and adolescents with primary hyperlipidemia	No change in oxLDL level	[153]
**Brazil nuts**				
	91 participants, 12 weeks, 13 g/day	Hypertensive and dyslipidemia subjects	Decrease in ox-LDL concentration; increase in glutathione peroxidase activity	[131]
	17 participants, 16 weeks, 15–25 g/day	Obese adolescents	Decrease in ox-LDL concentration	[130]
	50 participants, 12 weeks, 30 g/day	Metabolic syndrome	Reduction in oxidative stress	[92]
**Pecan**				
	44 participants, 4 weeks, 68 g/day	Older adults	Improving cholesterol profile and enhancing postprandial microvascular reactivity	[126]

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
