# Peer review of "The Cardioprotective Properties of Selected Nuts: Their Functional Ingredients and Molecular Mechanisms"

_foods, 2024, doi:10.3390/foods13020242_

Round 1

Reviewer 1 Report

Comments and Suggestions for Authors

Author Response

Thank you for your helpful comments. All of them have been taken into consideration when revising the manuscript. For example, I have added more information about carotenoids, copper, iron, magnesium, zinc, and sphingolipids:

“Some papers indicate that nuts are also plentiful sources of vitamins, including ascorbic acid, B1, B2, B3, B5, B6 and B9, carotenoids, including vitamin A  (almond – 2 µg/100 g, walnut - 21 µg/ml, pecan - 55 µg/ml, and hazelnut – 106 µg/100 g), and vitamin E.”

Moreover, a 30 g serving of cashew nuts has around 30 % of recommended daily intake of cooper. Hazelnuts, Brazil nuts, walnuts, pecans and pine nuts are also good alternative options [3,4]. In addition, macademia nuts, pine nuts, and cashews are source of iron [3,4]. It is an interesting that Brazil nuts (26 % of Recommended Dietary Allowances (RDA)), cashews (20 % of RDA), almonds (19 % of RDA), and pistachios (8% of RDA) are good source of magnesium [3,4]. Zinc was also found in various nuts, including pine nuts (8.7 mg/1 cup), hazelnuts (5.7 mg/1 cup), almonds (4.5 mg/1 cup), pecan (4.9 mg/1 cup), walnut (3.6 mg/1 cup), and pistachio (2.9 mg/1 cup) [3,4].”

Recently, sphingolipids, including sphingosine-1-phosphate (a bioactive metabolite that regulates diverse biological processes by binding to a family of G protein-coupled receptors or as an intracellular second messenger) were discovered in almonds and pistachios [3,4].”

I have added additional thoughts

  1. I have added new information: “It is an interesting that peanuts are botanically defined as legumes, however they have a similar nutrient composition and culinary use as tree nuts and are, therefore, usually included as nuts when estimating total nut intake [1-4].
  2. I have added new information: “It is an interesting that raw cashews contain urushiol, a toxin that causes a delayed allergic skin reaction similar to that caused by poison ivy. The poison is contained in the shells, so the kernels are perfectly safe [1-4].”
  3. I have added new information: “It is an interesting that pistachios contain chlorophyll, which is where they get their green color. In a way, they are the nutty equivalent of kale [1-4].
  4. Saturated fats. I have added new information: “It is an interesting that Brazil nuts, cashews and macadamia nuts have high level of saturated fatty acids that may be risk of CVDs [1-4].”
  5. I have added new information: “However, walnuts are actually the oldest tree food known to man. Some researchers claim that humans have been eating walnuts since as far 100,000 years ago [1-4].

Reviewer 2 Report

Comments and Suggestions for Authors

This paper reviewed the current state of knowledge regarding the functional ingredients of various nuts (almonds, Brazil nuts, cashew nuts, hazelnuts, macademia nuts, peanuts, pecan nuts, pine nuts, pistachios, and walnuts) and the effects with the possible action mechanisms related to their cardioprotective action. The data indicated that almonds, walnuts, and pistachios are good nut sources of bioactive ingredients with cardioprotective properties. There are some concerns (typos and others) as listed in the following.

*L67: 80.25 g/100?

*L70: (40.8 g/100 g, respectively)

**L80-81: Some studies indicate that their content may be decreased by higher temperatures and longer heating, and decreased by lower temperatures or shorter heating times [71].??

*L204: control knock-out mice-> control apoE-knockout mice

*L235: that ; for 12-week

L337: 40 µl/mL – 319 µL/mL

*L338: U937 cells - What kind of cell type?

L377: influence to -> influence on?

*L386: fam? cell formation > foam

*L418: Il-6 - interleukin-6 -> IL-6

*L491: Nutrition and Cancer -> Nutr Cancer

*L572: peanut allergen Ara h -> peanut allergen Ara h 1.

**L574: Nutrients 2023;15:1-16. -> Nutrients 2023, 15(4), 911; https://doi.org/10.3390/nu15040911 - 11 Feb 2023

*L677: Folia biol (Praha) > Folia Biol (Praha)

**L771: Table 1: PUFA in Pine nuts (34.1 g/100g) is higher than that of Pistachios (14.1 g/100g)

*L775: Table 2: It is better to shift the name of studied nuts (now in the center) to the right place under ‘Nuts’ and using Bold letter

Comments on the Quality of English Language

Minor editing of English language required

Author Response

This paper reviewed the current state of knowledge regarding the functional ingredients of various nuts (almonds, Brazil nuts, cashew nuts, hazelnuts, macademia nuts, peanuts, pecan nuts, pine nuts, pistachios, and walnuts) and the effects with the possible action mechanisms related to their cardioprotective action. The data indicated that almonds, walnuts, and pistachios are good nut sources of bioactive ingredients with cardioprotective properties. There are some concerns (typos and others) as listed in the following.

Thank you for your helpful comments. All of them have been taken into consideration when revising the manuscript.

*L67: 80.25 g/100?

Response: I have corrected. Now, it is: “80.25 g/100 g”.

*L70: (40.8 g/100 g, respectively)

Response: I have corrected. Now, it is: “Macademia nuts (58.9 g/100 g), and pecan nuts (40.8 g/100 g), respectively.”

**L80-81: Some studies indicate that their content may be decreased by higher temperatures and longer heating, and decreased by lower temperatures or shorter heating times [71].??

Response: I have added more information about it: “Some studies indicate that their content may be decreased by higher temperatures and longer heating, and decreased by lower temperatures or shorter heating times [71]. For example, phenolic components (total phenols, flavonoids, condensed tannins and phenolic acids) of almond kernels substantially lost in the initial phase; afterward these components gradually increased with roasting temperature and duration. Similar results also observed for their antioxidant activities [71]. Barreca et al. [2] report that roasting induces chemical and microstructural changes in almonds, favoring lipid oxidation and modulating antioxidant compound content; it also alters the protein profile and allergenic properties of nuts in general [72]. Interestingly, hot water blanching also influences the protein composition of nuts; however the effect depends on the species and conditions of processing [73].”

*L204: control knock-out mice-> control apoE-knockout mice

Response: I have corrected. Now, it is: “control apoE-knockout mice”.

*L235: that ; for 12-week

Response: I have corrected. Now, it is: “A recent study also found the consumption of 1.5 oz mixed tree nuts per day (for 12 weeks) to affect tryptophan metabolism, a factor in CVDs, in 56 overweight and obese subjects.”

L337: 40 µl/mL – 319 µL/mL

Response: I have corrected. Now, it is: “40 µl/mL of cells– 319 µL/mL of cells”.

*L338: U937 cells - What kind of cell type?

Response: I have added this information: “pro-monocytic, human myeloid leukaemia cell line”.

L377: influence to -> influence on?

Response: I have corrected. Now, it is: “influence on”.

*L386: fam? cell formation > foam

Response: I have corrected. Now, it is: “foam cell formation”.

*L418: Il-6 - interleukin-6 -> IL-6

Response: I have corrected. Now, it is: “IL-6”.

*L491: Nutrition and Cancer -> Nutr Cancer

Response: I have corrected. Now, it is: “Nutr Cancer”.

*L572: peanut allergen Ara h -> peanut allergen Ara h 1.

Response: I have corrected. Now, it is: ”peanut allergen Ara h 1.”

**L574: Nutrients 2023;15:1-16. -> Nutrients 2023, 15(4), 911; https://doi.org/10.3390/nu15040911 - 11 Feb 2023

Response: I have corrected. Now, it is: ”Nutrients 2023, 15(4), 911”.

*L677: Folia biol (Praha) > Folia Biol (Praha)

Response: I have corrected. Now, it is: ”Folia Biol”.

**L771: Table 1: PUFA in Pine nuts (34.1 g/100g) is higher than that of Pistachios (14.1 g/100g)

Response: I have corrected.

*L775: Table 2: It is better to shift the name of studied nuts (now in the center) to the right place under ‘Nuts’ and using Bold letter

Response: I have corrected.

Comments on the Quality of English Language

Minor editing of English language required

Response: I have corrected.